# The Prediction Analysis of Microarray 50 (PAM50) Gene Expression Classifier Utilized in Indeterminate-Risk Breast Cancer Patients in Hungary: A Consecutive 5-Year Experience

**DOI:** 10.3390/genes14091708

**Published:** 2023-08-28

**Authors:** Magdolna Dank, Dorottya Mühl, Annamária Pölhös, Renata Csanda, Magdolna Herold, Attila Kristof Kovacs, Lilla Madaras, Janina Kulka, Timea Palhazy, Anna-Maria Tokes, Monika Toth, Mihaly Ujhelyi, Attila Marcell Szasz, Zoltan Herold

**Affiliations:** 1Division of Oncology, Department of Internal Medicine and Oncology, Semmelweis University, H-1083 Budapest, Hungary; 2Department of Internal Medicine and Hematology, Semmelweis University, H-1088 Budapest, Hungary; 3Department of Pathology, Forensic and Insurance Medicine, Semmelweis University, H-1091 Budapest, Hungary; 4Department of Surgery, Transplantation and Gastroenterology, Semmelweis University, H-1082 Budapest, Hungary; 5Department of Radiology, Semmelweis University, H-1082 Budapest, Hungary; 6TritonLife Medical Center, H-1135 Budapest, Hungary; ujmisi@gmail.com

**Keywords:** breast carcinoma, subtype, immunohistochemistry, PAM50, gene expression, intrinsic subtype

## Abstract

Background: Breast cancer has been categorized into molecular subtypes using immunohistochemical staining (IHC) and fluorescence in situ hybridization (FISH) since the early 2000s. However, recent research suggests that gene expression testing, specifically Prosigna^®^ Prediction Analysis of Microarray 50 (PAM50), provides more accurate classification methods. In this retrospective study, we compared the results of IHC/FISH and PAM50 testing. We also examined the impact of various PAM50 parameters on overall survival (OS) and progression-free survival (PFS). Results: We analyzed 42 unilateral breast cancer samples, with 18 classified as luminal A, 10 as luminal B, 8 as Human epidermal growth factor receptor 2 (HER2)-positive, and 6 as basal-like using PAM50. Interestingly, 17 out of the 42 samples (40.47%) showed discordant results between histopathological assessment and the PAM50 classifier. While routine IHC/FISH resulted in classification differences for a quarter to a third of samples within each subtype, all basal-like tumors were misclassified. Hormone receptor-positive tumors (hazard rate: 8.7803; *p* = 0.0085) and patients who had higher 10-year recurrence risk scores (hazard rate: 1.0539; *p* = 0.0201) had shorter OS and PFS. Conclusions: Our study supports the existing understanding of molecular subtypes in breast cancer and emphasizes the overlap between clinical characteristics and molecular subtyping. These findings underscore the value of gene expression profiling, such as PAM50, in improving treatment decisions for breast cancer patients.

## 1. Introduction

According to the World Health Organization (WHO), breast cancer is the most common cancer among women worldwide, both in developed and developing countries. In 2020, an estimated 2.3 million new cases of breast cancer were diagnosed globally, accounting for 11.7% of all new cancer cases [1]. The lifetime risk of developing breast cancer for women is approximately 13% in developed regions [2].

Breast cancer subtypes were described by Dr. Charles M. Perou and his colleagues in a landmark study published in 2000 [3]. The study utilized gene expression profiling to identify distinct molecular subtypes of breast cancer. By analyzing the patterns of gene expression in breast tumors, the researchers identified four main subtypes: luminal A, luminal B, Human epidermal growth factor receptor 2 (HER2)-enriched, and basal-like. This classification of breast cancer subtypes based on gene expression patterns has since become widely used in research and clinical practice to guide treatment decisions [4].

Routine testing methods have been available to reflect the molecular subtypes of breast cancer since the 80s for hormone receptors [5]. These testing methods involve analyzing the expression levels of specific genes or proteins in the tumor tissue. The most common approach is to use immunohistochemistry (IHC) or in situ hybridization (ISH) techniques to assess the expression or amplification of certain biomarkers associated with each subtype [6].

Hormone receptor status [7,8] includes testing for estrogen receptor (ER) and progesterone receptor (PR) expression by immunohistochemistry. Positive expression of these receptors indicates that the tumor belongs to the luminal subtype (luminal A or luminal B). According to the 2013 St. Gallen criteria, 20% was the threshold for PR to distinguish between luminal A and B subtypes [9]. However, this was not confirmed at the following consensus meeting in 2015 [6]. Ki-67 is a protein marker that indicates the level of cell proliferation and is often used to differentiate between luminal A and luminal B subtypes. Luminal A tumors have a lower Ki-67 index, while luminal B tumors generally have a higher index and might also express HER2. HER2 overexpression or amplification is typically determined through IHC or ISH testing. Tumors with HER2 overexpression with absent ER or PR stains are classified as HER2-enriched subtype. Basal-like subtype is characterized by the absence of hormone receptors (ER, PR) and HER2 overexpression/amplification [9]. This subtype is determined by the absence of these markers in IHC testing and is called triple-negative breast cancer (TNBC) in routine. Further subtypes that are not part of the clinicopathological work-up emerged, such as claudin-low tumors [10].

While these histopathological testing methods provide valuable information about breast cancer subtypes, they are not the sole determining factors. Molecular profiling techniques, such as gene expression profiling using platforms like Oncotype DX and MammaPrint, can also provide comprehensive subtype information [11,12]. These tests analyze the expression patterns of multiple genes and provide a more detailed molecular classification of breast cancer. Oncotype DX (level 1A evidence) and MammaPrint (level 1B evidence) predict therapy response, while Prediction Analysis of Microarray 50 (PAM50) primarily was designed to provide subtyping information.

PAM50 is a gene expression-based assay used for molecular profiling of breast cancer, derived from the original identification method as determined by Perou et al. [3,13]. It is a molecular diagnostic tool that helps classify breast cancer into different intrinsic subtypes based on the expression patterns of 50 genes measured parallel with 8 housekeeping genes [14]. The PAM50 assay utilizes gene expression profiling, adapted to the NanoString method, to measure the activity levels of specific genes within a breast tumor sample [15,16]. The PAM50 assay provides valuable information about the molecular subtype of breast cancer, which can help guide treatment decisions and predict patient outcomes and is utilized in many countries in breast medical oncology. The risk of recurrence (ROR) score proved to be of prognostic value in early-stage luminal breast cancer (level 1B evidence) based on the TransATAC and ABCSG-8 trials [17,18,19,20]. Attempts have been made to utilize the PAM50 classifier in various clinical scenarios in prostate cancer as well [21,22,23].

Herein, we compared the gene expression-based classification with our routine histopathological characterization in a consecutive cohort of breast cancer patients with follow-up. This is possible when utilizing the PAM50 method of molecular tests. Our aim was dual: (i) to assess correlation of gene expression based and surrogate histopathological classifiers; and (ii) to evaluate the predictive capability of ROR score for early relapse in all breast cancer subtypes.

## 2. Materials and Methods

### 2.1. Patients and Study Design

A retrospective, single-center pilot study was conducted with the inclusion of female breast cancer patients, who attended the Division of Oncology, Department of Internal Medicine and Oncology, Semmelweis University, Budapest, Hungary between 2017 and 2021. Inclusion criteria for the study were age > 18 years, diagnosis of unilateral breast cancer either as primary or recurrent, an Eastern Cooperative Oncology Group (ECOG) performance status ≤2, and at least 2 years of follow-up data for survivors. Those patients’ tumors were considered indeterminate prognostically, and were analyzed herein, where (i) the hormone receptor expression was discordant by at least 60%; (ii) besides evidently expressing either ER or PR, expression was missing or lower than 20% considering the other hormone receptor; (iii) no hormone receptor was expressed, and HER2 expression was 2+ or 3+ while HER2 was not amplified by ISH; and (iv) no hormone receptor was expressed, but the Ki67 index was lower than 20%. Exclusion criteria included bilateral breast cancer, any previous or synchronous other malignancies, or any hematologic, systemic autoimmune, inadequately controlled thyroid, chronic kidney, or mental diseases. Only patients enrolled at the end of the study were followed up for a minimum two-year time interval; thus, the median follow-up period was 40.53 months.

The sample size for the study was estimated using the following assumptions. It was hypothesized that a 10–20% discrepancy might be between gene testing and conventional methods, which in the case of basal-like types might be even higher. An “effect size calculation in the chi-squared test for goodness of fit” was performed assuming the alternative hypothesis that 20–20–20–40% of the luminal A, luminal B, HER2-positive and basal-like tumors will be differently classified using the PAM50 gene expression profiling kit, respectively. Basal-like breast cancers can be more easily misclassified compared to the other three subtypes [24]. Using this method, a minimum of 39 total samples were estimated to obtain *p* < 0.05 significance with 80% statistical power.

### 2.2. Immunohistochemical Staining and PAM50 Testing of Tumor Samples

Breast cancer samples were obtained either from core biopsy or tumor resection. Both the routine IHC [25] and the PAM50 genetic testing were performed on the same samples. In accordance with international guidelines [25,26,27,28], the routine histopathological procedures included the IHC staining for ER, PR, HER2 and the proliferation marker Ki-67. IHC and fluorescence in situ hybridization (FISH) techniques were employed to analyze 4 μm thick tissue sections. IHC was performed to detect hormone receptor and HER2 expression using an automated immunostainer system (Ventana Benchmark XT, Roche Diagnostics, Mannheim, Germany). The antibodies used were ER (SP1), PR (1E2), and HER-2/neu (4B5). The hormone receptor status was evaluated using the Allred/Quick scoring system including the percentage of positively stained cancer cell nuclei, and the Ki67 labeling index was determined as the proportion of positively stained tumor cell nuclei. HER2 positivity was initially determined using IHC. FISH was initially performed on samples with IHC 2+ to assess HER2 gene amplification using Poseidon Repeat-Free probes. For further analysis and biologic understanding, HER2 1+ and 3+ cases were assessed in the indeterminate cohort. FISH results were evaluated based on the 2018 ASCO-CAP HER2 Test Guideline Recommendations [29]. Breast cancers were categorized into subtypes according to the St. Gallen International Expert Consensus on the Primary Therapy of Early Breast Cancer 2015 [6], including Luminal A-like, Luminal B-like (HER2-negative), Luminal B-like (HER2-positive), HER2-positive, and triple-negative subgroups. The threshold to distinguish between luminal A-like and B-like tumors was 20% for Ki67 [9]. Tumors with lower than 20% PR expression and or parallel HER2 expression were also classified as Luminal B.

In addition to the routine IHC staining, the Prosigna^®^ Breast Cancer Prognostic Gene Signature Assay (Veracyte Inc., South San Francisco, CA, USA) tests in NanoString FLEX Ncounter platform (NanoString, Inc., Seattle, WA, USA) were used to determine the gold-standard intrinsic breast cancer subtypes. This test was formerly called the PAM50 test, which abbreviation we will use throughout the paper. The panel examines the expression profiles of 50 tumor-specific genes [16,30,31]. In addition to the information about the molecular subtype of the tumor, the manufacturer also provides the so-called ROR score and a 10-year recurrence risk score [32,33]. The evaluation is completely independent and cannot be influenced by the staff conducting the examination. The ROR and 10-year recurrence risk scores are presented for exploratory purposes only for the patients with recurrent tumors, distant metastasis, and hormone receptor-negative tumors by IHC at baseline.

### 2.3. Clinical Characteristics

Complete clinicopathological data, including details of tumor removal surgeries, oncological therapies, etc. were collected. The tumor staging was given by histopathological examination of surgical primary tumor specimens and imaging studies. The 8th version of the American Joint Committee on Cancer (AJCC) grouping was used [34]. Oncologic treatment of patients was based on national and ESMO guidelines [35,36,37]. Due to the large number of possible combinations, the various types of possible oncological therapeutics were recorded only as dummy variables whether the patients received neoadjuvant, hormone, radio, chemo, or biological therapy. Overall (OS) and progression-free (PFS) survival was defined as the time between the PAM50 sampling and the death of any cause or any progression of the disease, respectively. The RECIST 1.1 guideline [38] was used to define disease progression. Patients without death/progression event(s) were right-censored. Follow-ups of patients using the hospital information system of Semmelweis University, whether a patient was still alive or deceased, were terminated on 31 May 2023.

### 2.4. Statistical Analysis

Statistical analysis was performed in the R for Windows version 4.3.0 environment (R Core Team, 2023, Vienna, Austria). Cohen’s kappa (κ) was calculated using the R package “psych” (version 2.3.6). Patient survival was evaluated using the Cox regression model (R package “survival”, version 3.5-5). As some samples were obtained from recurrent tumors, while others from primary ones, all survival analyses were performed by adjusting the baseline hazards (also known as stratification) to eliminate the effect that was present due to the primary-recurrent difference; thus, the biasing effect of metastatic and recurrent tumors were omitted from the survival analyses. *p* < 0.05 was considered statistically significant. If not stated otherwise, continuous, count, and survival data were expressed as the mean ± standard deviation, the number of observations (percentage), and the hazard ratio (HR) with a 95% confidence interval (95% CI), respectively. Survival curves were drawn with the “survminer” R-package (version 0.4.9).

## 3. Results

In this study, 42 patients were included, with an average age of 56.78 ± 14.32 years. Out of these patients, 37 had primary tumors, while 5 had recurrent tumors. Among the 42 patients, 5 (11.90%) died during the study (Appendix A), and a total of 10 (23.81%) progression events were recorded, including the 5 deaths. Specifically, four deaths and three progressions were observed in patients with primary tumors, while one death and two progressions were observed in patients with recurrent tumors. The clinicopathological parameters of the study participants are summarized in Table 1. Respectively, five and seven patients received no and a single treatment type only, while the rest of the patients received at least two types of adjuvant/metastatic series. As the histology samples were obtained either from core biopsy or from surgical resection, it was investigated whether it influenced patient survival. No differences between the two sample-obtaining techniques could be justified (OS: *p* = 0.9330; PFS: *p* = 0.2170). No difference could be verified in the case of tumor size (*p* = 0.4596), lymph nodes (*p* = 0.0637), nor for distant metastasis (*p* = 0.1264).

### 3.1. PAM50 Results

In addition to routine histopathological procedures, the molecular subtypes of the 42 study participants were determined using PAM50 genetic tests. Based on these tests, 18 patients were classified as luminal A, 10 as luminal B, 8 as HER2-positive, and 6 as basal-like subtypes. The manufacturer of PAM50 also offered to give ROR and 10-year recurrence risk scores, the results of which are summarized in Table 2. According to the ROR scores, 8 patients were classified as low-risk, 13 patients as moderate-risk, and 20 patients as high-risk.

As detailed above, a total of 10 progression events were observed, of which 5 deaths occurred. The effects of PAM50 parameters were investigated for whether they have any effects on OS and/or PFS. All survival models were refined using baseline hazard adjustment to remove the biasing effect of primary vs. recurrent tumors. None of the PAM50 parameters had any significant effect over OS. A significant difference between the PFS of patients with the luminal A and HER2-positive (*p* = 0.0149) and basal-like (*p* = 0.0110) subtypes was observed (Figure 1). Moreover, the higher the 10-year recurrence risk score was, the shorter the PFS time was (increase in risk for shorter survival per unit: 1.0529; *p* = 0.0201). The complete list of PFS analyses is listed in Table 3. It was also investigated whether neoadjuvant therapy changes the above detailed results; the same was found in those models as well.

### 3.2. Comparison of PAM50 and Immunohistochemical Staining Results

In addition to PAM50 testing, the same samples were also analyzed by a pathologist using IHC staining methods. The estrogen, progesterone, HER2, and Ki-67 IHC percentage results, on which the pathological results were based, can be read in Table 4 and Table 5. A total of 15, 18, 8, and 1 of the tumors were classified as luminal A, luminal B, HER2-positive and as triple-negative breast cancer, based on the IHC and FISH testing, respectively. By comparing the two approaches directly, 17 (40.48%) of the samples were differently characterized by the pathologist(s), compared to that of the PAM50 genetic test results. Based on IHC and FISH testing, out of the 17 misclassified tumors, 2 were classified as luminal A, 11 as luminal B, 3 as HER2-positive, and 1 as basal-like. Specifically, five samples were determined as luminal B using the routine methods, which should have been classified as luminal A, according to PAM50 testing. Additionally, two samples were classified as luminal A and one as triple-negative based on routine methods, whereas these three samples were actually luminal B tumors according to PAM50 testing. Three samples were classified as luminal B, instead of the PAM50 determined HER2-positive. Notably, all six basal-like tumors were characterized differently by IHC/FISH: three were classified as HER2-positive and three as luminal B (Figure 2). Using Cohen’s κ, a moderate agreement could be justified between the IHC and PAM50 classifications (Cohen’s κ: 0.43; 95% confidence interval: 0.23–0.62).

## 4. Discussion

In this study, 42 patients were included, whose breast tumors were subjected both to immunophenotypical analysis and gene expression-based classification of intrinsic subtypes. The concordance between the routine surrogates and the multigene standard method was modest, with a 59.5% concordance. It is known that PAM50 molecular subtyping can oftentimes yield different results compared to routine analysis methods in breast cancer. Several studies have compared the results of PAM50 testing with routine analysis methods and have reported discordant findings in a subset of cases. These discrepancies can arise due to differences in the preanalytical–analytical steps and the intrinsic properties of breast cancer biology. For example, PAM50 considers the expression levels of a larger number of genes associated with breast cancer subtypes, whereas routine analysis methods rely on the assessment of four markers: receptor expression and Ki67 index. A study published in 2015 evaluated the concordance between PAM50 subtyping and routine IHC-based subtyping in a large cohort of breast cancer patients [20]. The authors found that approximately 25% of cases showed discordant subtyping results. Another relatively recent study analyzed a similar question in a Sub-Saharan African setting: the luminal A tumors were concordant in 81%, luminal B in about 53%, HER2-enriched in 85%, and TNBC/basal-like tumors in 88% [39]. In an earlier comparison, it was noted that hormone receptor expression was detected in luminal gene expression groups in about 92%, while only 75% of hormone-receotor-negative tumors belonged to the HER2-enriched and basal-like intrinsic subgroups [40]. Erber et al. performed an assessment in a German-Austrian real-world setting, where they found that the concordance was fair to moderate using Cohen’s κ as an index of pairwise agreement between PAM50 subtypes and local IHC based assessments with a range of 0.34–0.45 [41]. In our current analysis, Cohen’s κ was 0.43, which corresponds to the most accurate value of the coordinating center (κ = 0.44) in the above study [41]. This highlights the potential for divergent classification when comparing PAM50 with routine analysis. By adding histological grade, the correlation was improved [41]. Our patient cohort was special in means of indeterminate (or discrepant) routine receptor expressions and Ki67 index; therefore, a higher discordance between gene expression-based and protein-based assessment was expected.

The findings we described, regarding the association between molecular subtypes of breast cancer and the expression of hormone receptors and HER2, are consistent with previous studies [42,43,44]. Luminal A subtype: Luminal A tumors are characterized by high levels of hormone receptor expression (median ER: 100% and PR: 82%). Additionally, they typically exhibit low proliferation, as indicated by the Ki67 index, with a median of 10% and a maximum of 25%. Luminal B and HER2-enriched subtypes: These subtypes are characterized by lower expression of progesterone receptor (PR) and variable expression of ER. From a developmental perspective in the mammary gland, these cancers can be considered as part of a continuum rather than distinct entities that can be clearly differentiated [45]. In luminal B tumors, there is often low PR expression (median: 10.5%) while estrogen receptor (ER) expression is preserved (median: 99.5%). In the HER2-enriched group, hormone receptor expression is minimal, and HER2 expression was evident only in the largest proportion of cases (62.5%). Basal-like subtype: Basal-like tumors often exhibit aberrant expression of hormone receptor and/or HER2. It is noteworthy that all basal-like tumors in our study expressed some form of hormone receptor or displayed HER2, which may reflect their unique phenotype and contribute to differences in predicted survival. Some basal-like tumors were HER2-positive, while others showed high levels of hormone receptor expression (e.g., ER: 90%). In a study, about half of ER-low (<10%) breast tumors were found to be basal-like, and the other half behaved as luminal based on gene expression classification [46].

According to the Stockholm 2019 Guidelines, low-grade and large-size tumors, high-grade and small-size tumors, and intermediate-grade/risk breast cancers are to be tested by PAM50 in Sweden [47]. The Danish Breast Cancer Cooperative group collected and tested a consecutive hormone receptor-positive, HER2-negative cohort with PAM50, and concluded that the 10-year prognostic prediction is helpful in identifying those patients who can spare chemotherapy [48]. In France, the availability of PAM50 test results significantly enhanced the confidence of attending physicians in formulating adjuvant treatment determinations. This, in turn, precipitated a notable 18% alteration in chemotherapy treatment strategies [49]. The application of PAM50 testing elicited a discernible reduction in patient anxiety levels and a concomitant improvement in quantifiable metrics pertaining to health-related quality of life within the cohort undergoing adjuvant therapy. Moreover, the observation of a 25% incongruity between the outcomes of the PAM50 test and IHC subtyping serves to underscore the critical importance of molecular testing for the optimization of indications guiding systemic therapeutic interventions in the context of early-stage breast cancer. The inaugural investigation applying the PAM50 test to women in the Middle East with early-stage breast cancer found a notable 23.4% revision in adjuvant treatment stratagems [50]. The PAM50 test is endorsed by the German Association of Gynecologic Oncology (AGO), the European Society for Medical Oncology (ESMO), the American Society of Clinical Oncology (ASCO), and the National Institute for Health and Care Excellence (NICE) in the United Kingdom, as stipulated in their respective guidelines.

It is important to note that PAM50 molecular subtyping has been shown to provide valuable prognostic and predictive information, and it is considered a more comprehensive and precise method for classifying breast cancer subtypes. Here, the 10-year risk of recurrence was highest in luminal B and HER2-enriched subgroups, relatively lower in the basal-like subgroup, and lowest—as expected—in luminal A tumors. In a resource-constrained setting, PAM50 testing is often considered complementary to routine analysis and can provide additional insights into the molecular characteristics of breast cancer. Primarily, subtyping can give further clues when a given patient’s tumor is non-responsive to treatment.

With all this in mind, we can hypothesize to test breast cancer samples for PAM50 classifier when the routine characterization or clinical data yield:ER and/or PR below 35% for accurate breast cancer intrinsic subtyping.Those samples where the tumor response to a given therapy fails (e.g., residual disease after or progression on neoadjuvant therapy); testing these would be a rational approach.Grade 1 tumors that are ≥pT2.Grade 3 tumors that are ≤pT1c.An intermediate category with other tests to provide additional support for therapeutic decisions.An indication for extended hormone therapy after 5 years of hormone therapy.

### Limitations of the Study

It is important to note that these observations are based on the limited data provided, and further studies involving larger cohorts are necessary to confirm these findings and explore their clinical implications. There were some discrepancies between staging and surgery details, and only limited data were available about the oncological treatments of the study participants. The small sample size prevented us from creating multivariate survival models. The cohort is “indeterminate clinical risk”, which obviously has an effect on the resulting data, mostly due to pre-selection bias. Ki67 assessment can suffer from interobserver variability, which has long been recognized by medical professionals and consensus committees. The follow-up time was relatively short, although we attempted to analyze the outcome after a minimum of 24 months following diagnosis for early breast cancer, as this already reflects the peak of early recurrences of high-risk tumors. Taken together, we draw conclusions based on the Hungarian setting of the initial use of PAM50 intrinsic gene expression classifier. A further limitation of the study was the lack of a control group. Moreover, the heterogenous follow-up times of the study participants also included some bias. Therefore, it would be worthwhile to repeat the study with a larger number of cases and a longer follow-up.

## 5. Conclusions

These findings support the existing knowledge of molecular subtypes in breast cancer and their corresponding hormone receptor and HER2 expression patterns. While our study had limitations, we observed that routine clinical characteristics and gold-standard molecular subtyping largely aligned but were not completely overlapping. Our aim was to characterize tumors with indeterminate clinical risk and gain further insights into breast cancer biology, enabling more informed therapeutic decisions through gene expression profiling. It is important to note that the discrepancy between IHC and PAM50 is not exclusive to our study and has been observed in other investigations as well. The PAM50 test is endorsed by the German Association of Gynecologic Oncology (AGO), the European Society for Medical Oncology (ESMO), the American Society of Clinical Oncology (ASCO), and the National Institute for Health and Care Excellence (NICE) in the United Kingdom, as stipulated in their respective guidelines.

## Figures and Tables

**Figure 1 genes-14-01708-f001:**
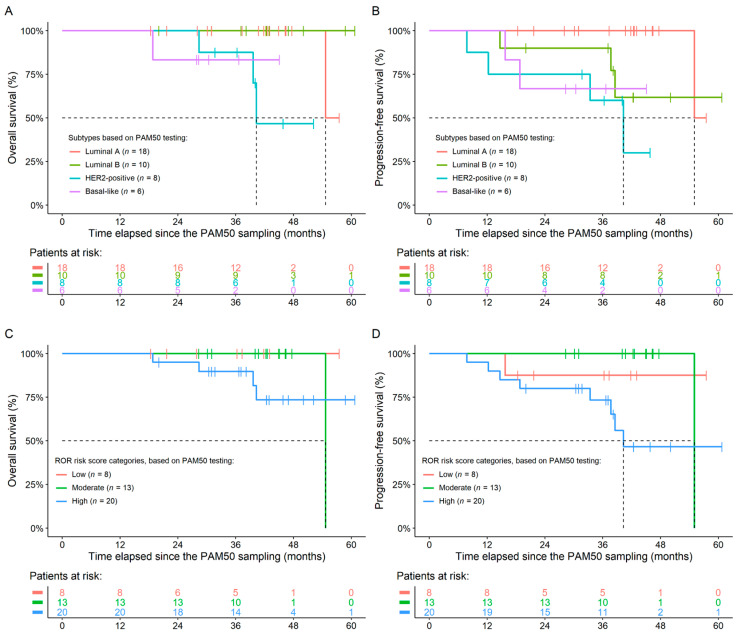
Overall (**A**,**C**) and progression-free (**B**,**D**) survival results of the study participants. Grouping was performed based on the PAM50 subtypes (**A**,**B**) and PAM50 risk of recurrence (ROR) scores (**C**,**D**). Significant differences between the survival times of any patient groups were observed only in the case of PFS.

**Figure 2 genes-14-01708-f002:**
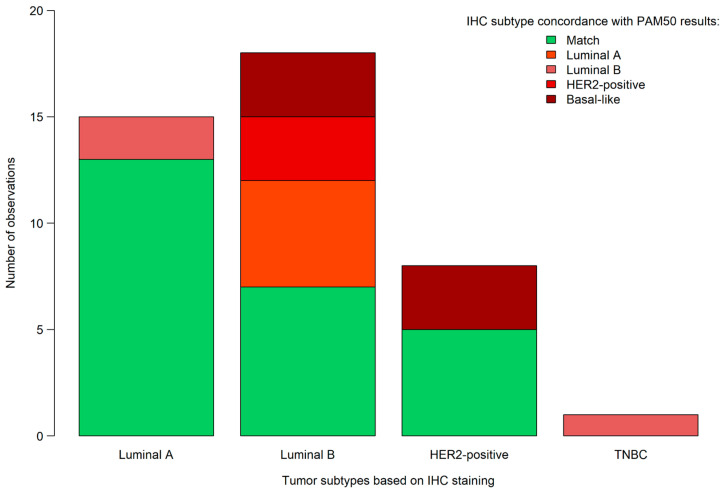
Comparison of breast cancer subtypes based on PAM50 testing and immunohistochemistry (IHC). TNBC: triple-negative breast cancer.

**Table 1 genes-14-01708-t001:** Clinicopathological features of the analyzed cohort. The units of continuous and count data are the mean ± standard deviation and the number of observations (percentage), respectively. The discrepancy between staging and surgery details occurred due to the fact that some of the patients were not operated on. Where available, the staging of these patients was obtained from imaging studies (cTNM).

Clinical Characteristics	Study Participants (*n* = 42)
Age (years)	56.78 ± 14.32
Sample obtained from	
- Primary tumor	37 (88.10%)
- Recurrent tumor	5 (11.90%)
Type of specimen	
- Core biopsy	10 (23.81%)
- Surgical resection	32 (76.19%)
Histology of the tumor	
- Invasive breast carcinoma NST	31 (73.81%)
- Invasive lobular carcinoma, classical	5 (11.90%)
- Primary squamous cell carcinoma	1 (2.38%)
- Papillary carcinoma	1 (2.38%)
- Pleomorphic lobular carcinoma	1 (2.38%)
- Mixed type	3 (7.14%)
Nottingham histologic grade ^1^	
- 1	7 (18.2%)
- 2	19 (51.35%)
- 3	11 (29.72%)
TNM staging [34] ^1^	
- Tumor size (T1:T2:T3:T4)	18:13:8:1 (42.86%:30.95%:19.05%:2.38%)
- Lymph node metastasis (N0:N1:N2:N3)	19:17:3:1 (45.24%:40.48%:7.14%:2.38%)
- Distant metastasis (M0:M1)	37:2 (88.10%:4.76%)
Size of the tumor (mm)	27.00 ± 22.82
Surgical procedures	
- Breast-conserving surgery	18 (42.86%)
- Total mastectomy	19 (45.24%)
- Axillary block dissection (ABD)	15 (35.71%)
- Sentinel lymph node biopsy (SNB) - ABD following SNB	19 (45.24%) 6 (14.29%)
Neoadjuvant therapy	12 (28.57%)
Adjuvant therapy	
- Endocrine therapy	30 (71.43%)
- Radiotherapy	26 (61.90)
- Chemotherapy	18 (42.86%)
- Targeted therapy	9 (21.43%)
Median progression-free survival (months)	55.03 (95% CI: 55.03–NA) ^2^

^1^ Stage T, N, and M and Nottingham-grade data were missing for 2, 2, 3 and 5 patients, respectively. ^2^ The upper 95% CI cannot be calculated due to the lower number of events. CI: confidence interval; NST: no special type.

**Table 2 genes-14-01708-t002:** Risk of recurrence (ROR) and 10-year recurrence risk scores of the study participants.

PAM50 Subtypes	Min.	25%	Median	75%	Max.	Mean	SD
ROR score							
Luminal A (*n* = 18)	1	17	38.33	42	47	30.36	14.02
Luminal B (*n* = 10)	54	66.75	73	81.33	86	72.80	10.24
HER2-positive (*n* = 8)	16	53.38	66	69	80.67	59.15	20.33
Basal-like (*n* = 6)	9	18.25	33	59	65	36.83	24.38
10-year recurrence risk score							
Luminal A (*n* = 18)	3	4	7	8	43	10.20	10.31
Luminal B (*n* = 10)	12	28.17	31.84	34.5	49	31.33	10.86
HER2-positive (*n* = 8)	4	15.75	20.5	26.25	43	21.08	11.96
Basal-like (*n* = 6)	4	10.25	14	25.25	43	18.83	14.36

Max: maximum; Min: minimum; SD: standard deviation.

**Table 3 genes-14-01708-t003:** Progression-free survival results.

Parameter	HR	95% CI	*p*-Value
PAM50 subtypes			
- Luminal A (ref.) vs. Luminal B	5.7420	0.5963–55.2900	0.1304
- Luminal A (ref.) vs. HER2-positive	20.7140	1.8078–237.3400	0.0149
- Luminal A (ref.) vs. Basal-like	31.8900	2.2086–460.4600	0.0110
- Luminal B (ref.) vs. HER2-positive	3.6076	0.5861–22.2060	0.1660
- Luminal B (ref.) vs. Basal-like	5.5541	0.6709–45.9790	0.1120
- HER2-positive (ref.) vs. Basal-like	1.5395	0.2385–9.9378	0.6502
ROR score	1.0181	0.9878–1.0490	0.2450
ROR risk categories			
- Low (ref.) vs. Moderate	0.6004	0.0369–9.7590	0.7200
- Low (ref.) vs. High	4.0626	0.4984–33.1190	0.1900
- Moderate (ref.) vs. High	6.7661	0.8357–54.7800	0.0732
10-year recurrence risk score	1.0529	1.0080–1.1000	0.0201

CI: confidence interval; HR: hazard rate; ref: reference group.

**Table 4 genes-14-01708-t004:** Estrogen-, progesterone- and Ki-67-positivity results of the study participants, based on the immunohistochemical staining performed by the pathologist(s).

PAM50 Subtypes	Min.	25%	Median	75%	Max.	Mean	SD
Estrogen receptor positivity (%)							
Luminal A (*n* = 18)	10	95.75	100	100	100	92.39	21.26
Luminal B (*n* = 10)	1	92	99.5	100	100	87.80	30.76
HER2-positive (*n* = 8)	0	0	0	15	90	18.75	35.63
Basal-like (*n* = 6)	0	0	0	22.5	90	20.00	36.33
Progesterone receptor positivity (%)							
Luminal A (*n* = 18)	0	36.25	82.5	93.75	100	62.83	38.38
Luminal B (*n* = 10)	0	1	10.5	50	100	30.10	40.75
HER2-positive (*n* = 8)	0	0	0	0	40	5.00	14.14
Basal-like (*n* = 6)	0	0	35	77.5	95	40.83	45.43
Ki-67 proliferation marker ratio (%)							
Luminal A (*n* = 18)	1	5	10	15	25	10.00	7.00
Luminal B (*n* = 10)	10	15	17.5	20	40	19.50	9.26
HER2-positive (*n* = 8)	10	25	25	30	80	34.00	26.79
Basal-like (*n* = 6)	30	60	70	80	80	64.00	20.74

Max: maximum; Min: minimum; SD: standard deviation.

**Table 5 genes-14-01708-t005:** HER2-positivity results of the study participants, based on the immunohistochemical staining.

PAM50 Subtypes	0	1+	2+	3+
Luminal A (*n* = 18)	11	2	5	0
Luminal B (*n* = 10)	5	2	3	0
HER2-positive (*n* = 8)	0	1	2	5
Basal-like (*n* = 6)	2	0	0	4

## Data Availability

The datasets used and/or analyzed during the current study are available from the corresponding author upon reasonable request.

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
