# Peer review of "The Prediction Analysis of Microarray 50 (PAM50) Gene Expression Classifier Utilized in Indeterminate-Risk Breast Cancer Patients in Hungary: A Consecutive 5-Year Experience"

_genes, 2023, doi:10.3390/genes14091708_

Round 1

Reviewer 1 Report

The article is adequately designed, it has main necessary elements, but need major revision about bellow proposed questions and after corrections, article require re-evaluation.

Title say ’’a consecutive 5-year experience’’ and it is not clear to what time frame is those 5 year regarding to? Patients were received oncology treatment between 2019-2021, and follow up lasted until May 31. 2023.

In introduction to many space is dedicated to breast cancer epidemiology and subtype classification (and determination methods), facts that are well known to researches in this particular field.

In introduction: ICH or FISH for assess amplification of certain biomarkers, but later on, this claim in further explanations show it to be inadequate.

It would be useful that molecular profiling techniques (Oncotype DX, MammaPrint and PAM50) were explained with its pros and cons and why is authors choice last listed one.

Does PAM50 helps only in breast medical oncology?

Study was retrospective? And patients were followed for two years? Please, explain this discordance.

Does ECOG PS > 0 suggest that patients with ECOG PS 0 were not included in the study and why?

Also, in patient characteristic, breast cancer acceptable for this study might be primary or recurrent? Was any difference between their subtypes? Place of recurrence and DFS (disease free survival) is not defined anywhere in the study.

Breast cancer samples were obtained from core needle biopsy (10 patients in total) or tumor resection, indicating that based on numbers in Tables lower in the article, all patients within groups did not received same treatment. Or that patients started surgical oncology treatment without histopathology confirmation. Please, clarify.

Start time was defined as time of PAM50 sampling, suggesting that sampling was taken at the same time as malignancy confirmation? Another question arising from this claim – are for patients with recurrent cancer in this study, follow up started from PAM50 sampling?

Please clarify sentence ‘’patients without death/progression event(s) were right-censored’’.

Please, how survival analyses were performed by adjusting the baseline hazards?

In results is not explained which patients died or relapsed during follow up, their TNM stage, tumor biology (subtype) or PAM50 score?

In Table 1, is stated that 15 patients had axillary dissection and that 17+3 patients had N1 or N2 disease. Please, explain this.

In clinical characteristics is stated that ‘’Due to the large number of possible combinations, the various types of possible oncological therapeutics were recorded only as dummy variables whether the patients received neoadjuvant, hormone, radio, chemo, or biological therapy’’ than, its needlessly to divide patients into groups based on oncology treatment they received, moreover, significant number of study population had multidisciplinary oncology treatment.

Can median PFS can be 55.03 month if all patients were followed for two years?

In the article is stated that none of PAM50 parameters affected OS? Was any difference between disease stage observed?

Please, explain sentence’ Moreover, the higher the 10-year recurrence risk score was, the shorter the survival time was.’’ To which group of patients this claim applies?

Are those two groups comparable: hormone receptor positive (ref.) vs. negative (p 0.0085), if in HR negative are all TNBC and HER2+ patients, with indetermine risk and undefined disease stage?

Can German real-world study can be defined in numbers instead: the concordance was fair to moderate?

‘’ Basal-like tumors often exhibit aberrant expression of HR and/or HER2. It is noteworthy that all basal-like tumors in your study expressed some form of HR or displayed HER2, which may reflect their unique phenotype and contribute to differences in predicted survival’’. Please, explain pronoun YOUR in this sentence.

Indetermined clinical risk is not defined in article.

Weather PAM50 testing is recommended for some particular group of patients?

Statistical power of study is missing. Can You please explain how you determine that 42 patient are adequate study sample?

In ‘’funding’’ only APC was explained. Can you please provide source of Prosigna kits? Is that mean that testing is widely available for all patients and if not, who decides which patients undergo testing?

Outdated references – more than 50% of references need to be within last five years.

English language require moderate editing.

English language require moderate editing.

Reviewer 2 Report

The retrospective study titled "The PAM50 gene expression classifier utilized in indeterminate-risk breast cancer patients in Hungary: a consecutive 5-year experience" analyzed 42 unilateral breast cancer samples to evaluate the accuracy of the Prosigna® PAM50 gene expression testing compared to traditional immunohistochemical staining (IHC) methods. The researchers found a 40% discrepancy between the two methods, with PAM50 identifying various molecular subtypes that were misclassified by IHC. Furthermore, hormone receptor-positive tumors and patients with higher 10-year recurrence risk scores were found to have shorter survival rates. Despite the study's limitations, including a small sample size, pre-selection bias, and a short follow-up period, the findings underscored the potential of gene expression profiling in improving treatment decisions for breast cancer patients.

I have found several critical weaknesses and limitations in the presented study. These are outlined below:

·        The authors based their study on a small cohort of only 42 unilateral breast cancer samples, which is not sufficient to generalize the results. The statistical power of the analysis is therefore limited and makes it challenging to determine the accuracy of the PAM50 gene expression classifier. A larger study population would have provided more robust and reliable results.

·        The absence of a control group in the study raises the concern of potential confounding factors that could have influenced the study's results. A control group would have facilitated a more definitive interpretation of the PAM50 test's efficacy.

·        The authors highlight a 40% discordance rate between immunohistochemistry and PAM50 in their study but fail to provide an adequate explanation for such a high discrepancy. This calls into question the reliability of either method, which the authors fail to address appropriately.

·        The relatively short follow-up period in this study might have affected the reliability of the findings related to survival rates. A longer follow-up would have facilitated a more comprehensive understanding of the impact of the PAM50 classifier on overall survival and progression-free survival.

·        The study focuses on an "indeterminate clinical risk" cohort which, as the authors have acknowledged, introduces a pre-selection bias. This substantially limits the generalizability of the study findings to other breast cancer populations.

·        The authors propose using PAM50 in several specific clinical scenarios but offer no data from their study to support these recommendations. For instance, they have not shown that PAM50 would be beneficial in tumors that progress during neoadjuvant treatment or are residual after such treatment.

·        The manuscript lacks the provision of sufficient data to support the conclusions drawn. There's an absence of confidence intervals for hazard ratios which are crucial for understanding the precision of these estimates. More rigorous statistical analyses and interpretation of data could have strengthened the study's conclusions.

In conclusion, while the manuscript attempts to address an important aspect of breast cancer diagnosis, it suffers from significant limitations that hamper its validity and potential impact. Without addressing these limitations, the current version of the manuscript does not adequately support the proposed use of the PAM50 gene expression classifier.

Reviewer 3 Report

This is a retrospective study comparing results of IHC/FISH and Prosigna PAM50 testing on 42 breast cancer samples. Overall, the manuscript is well-written and presents real-world findings. However, several larger studies derived from prospective cohorts, clinical trial participants, and real-world settings have previously examined this question and I am unclear the added value of this small study to the existing literature. The manuscript requires major revisions as outlined below and a convincing argument as to the value of this study if it is to be suitable for publication in this journal.

1.     The most significant methodological issue with this manuscript is the inclusion of patients in ROR analyses for which the test has not been validated. Specifically, it is noted there were 2 patients with distant metastasis and 5 patients with recurrent breast cancer, in which ROR and 10-year recurrence risk scores are inappropriate. Patients with no ER or PR expression and those with HER2-positive tumors by IHC should also be excluded from ROR and 10-year recurrence risk calculations, as the use of Prosigna for prognostic purposes has only been studied in patients with hormone receptor-positive HER2-negative tumors.

2.     Likewise, it is inaccurate to include metastatic patients and patients with recurrent in survival analyses according to PAM50 subtypes because Prosigna prognostic performance has not been validated in these subgroups.

3.     A multivariate survival analysis adjusting for clinicopathologic factors, treatment, and PAM50 classification would be more robust in terms of delineating the prognostic value of PAM50 subtypes. If the sample is too small for such an analysis, this should be acknowledged as a limitation.

4.     Did a single pathologist perform all IHC scoring? Please specify.

5.     What Ki-67 cutoff was used to differentiate luminal A versus B tumors by IHC? The manuscript cited provides a range and thus it is unclear what was used for this study.

6.     With reference to the discussion point, even in non-resource constrained settings PAM50 testing is infrequently used given the cost and questions regarding additional benefit to other genomic assays like Oncotype DX that are more widely used.

7.     I disagree with some of the proposed indications for PAM50 classifier. More importantly, the results of this small study as currently described and current discussion regarding prior literature do not justify any of these proposed indications. This section should either be removed or better rationalized.

8.     Other limitations of this study include the interobserver variability of Ki-67 and how this may have impacted the discordant results between IHC surrogates subtypes and PAM50 subtypes.

Round 2

Reviewer 1 Report

The authors adequately addressed most of the comments and made corrections accordingly in the paper, but some additional minor revisions are needed to ensure the paper is accepted for publication.

Suggestion about using PAM50 was to breast MEDICAL oncology, as a part of multidisciplinary breast cancer treatment, not including other malignancies. For example, does PAM50 helps in decision making for breast surgery? 

The discrepancy between staging and surgery details must be included as study weakness (or disadvantage) in discussion.

Authors answer: “5 and 7 patients received no and a single treatment type only, while the rest of the patients received at least 2 types of adjuvant/metastatic treatments” is corrected in article, but should also be mentioned as article disadvantage in discussion.

Minor editing

Author Response

Budapest, August 23, 2023.

Dear Reviewer,

On behalf of my fellow authors, first of all, I would like to thank you for your attention and opinion on our original article entitled "The PAM50 gene expression classifier utilized in indeterminate-risk breast cancer patients in Hungary: a consecutive 5-year experience". Here we provide answers to the questions and critiques our first Reviewer raised.

The authors adequately addressed most of the comments and made corrections accordingly in the paper, but some additional minor revisions are needed to ensure the paper is accepted for publication.
Suggestion about using PAM50 was to breast MEDICAL oncology, as a part of multidisciplinary breast cancer treatment, not including other malignancies. For example, does PAM50 helps in decision making for breast surgery? 
To our knowledge, no previous study investigated whether PAM50 testing at the time of the diagnosis can provide enough evidence for therapeutic decision. Indeed, it is a very intriguing question, however, to answer the question that whether a patient should be operated or treated with neoadjuvant chemo based on PAM results, cannot be answered neither by our study, nor by previous ones from the literature.

The discrepancy between staging and surgery details must be included as study weakness (or disadvantage) in discussion.
Thank you. It was added to the “Limitations of the study” section.

Authors answer: “5 and 7 patients received no and a single treatment type only, while the rest of the patients received at least 2 types of adjuvant/metastatic treatments” is corrected in article, but should also be mentioned as article disadvantage in discussion.
Thank you. It was added to the “Limitations of the study” section.

Yours sincerely,

            Dr. A. Marcell Szasz
            Semmelweis University
            Department of Internal Medicine and Oncology

Reviewer 2 Report

The reviewers have addressed all my comments and revised the manuscript accordingly. The manuscript can be accepted for publication.

Author Response

Budapest, August 22, 2023.

Dear Reviewer,

On behalf of my fellow authors, first of all, I would like to thank you for your attention and opinion on our original article entitled "The PAM50 gene expression classifier utilized in indeterminate-risk breast cancer patients in Hungary: a consecutive 5-year experience".

The authors have addressed all my comments and revised the manuscript accordingly. The manuscript can be accepted for publication.
Thank you for your kind comment.

Yours sincerely,

            Dr. A. Marcell Szasz
            Semmelweis University
            Department of Internal Medicine and Oncology

Reviewer 3 Report

Thank you for revising the original manuscript. I have a few follow up comments based on the changes that have been made. Specifically,

·      As the authors state, ROR and 10-year recurrence risk scores may be used in luminal breast cancer subtypes according to level 1B evidence. Thus, it should be clearly stated in the methods that the ROR and 10-year recurrence risk scores are presented for exploratory purposes only for the patients with recurrent tumors, distant metastasis, and hormone receptor-negative tumors by IHC at baseline.

·      The authors state that they agree patients with metastatic disease or recurrent tumors should be omitted from survival analyses according to PAM50 subtypes, but I do not see where they have clearly stated this in the methods that these patients were omitted from such calculations.

·      The authors state that PAM50 could hypothetically be used to establish an indication for extended endocrine therapy. Please provide some evidence in the discussion regarding Prosigna in late recurrence risk prediction to justify this statement.

Author Response

Budapest, August 22, 2023.

Dear Reviewer,

On behalf of my fellow authors, first of all, I would like to thank you for your attention and opinion on our original article entitled "The PAM50 gene expression classifier utilized in indeterminate-risk breast cancer patients in Hungary: a consecutive 5-year experience". Here we provide answers to the questions and critiques our third Reviewer raised.

Thank you for revising the original manuscript. I have a few follow up comments based on the changes that have been made. Specifically,
As the authors state, ROR and 10-year recurrence risk scores may be used in luminal breast cancer subtypes according to level 1B evidence. Thus, it should be clearly stated in the methods that the ROR and 10-year recurrence risk scores are presented for exploratory purposes only for the patients with recurrent tumors, distant metastasis, and hormone receptor-negative tumors by IHC at baseline.
Thank you for pointing this out. As we know, based on the two prospective cited clinical trial, this is proven in the luminal tumours, and ROR and 10-year risk of recurrence can be utilised in this cohort in the routine setting.

The authors state that they agree patients with metastatic disease or recurrent tumors should be omitted from survival analyses according to PAM50 subtypes, but I do not see where they have clearly stated this in the methods that these patients were omitted from such calculations.
We have clarified further this question to make its understanding more clear.

The authors state that PAM50 could hypothetically be used to establish an indication for extended endocrine therapy. Please provide some evidence in the discussion regarding Prosigna in late recurrence risk prediction to justify this statement.
As the clinical trials which have provided evidence for the long-term predictive capacity of the PAM50 classifier, over a 10-year span, the establishment of decision over an extended (over 5 years) hormone therapy is justified by this method. We agree, this has not been a head to head comparison and aim of the trials, this, it is stated carefully and this is also mentioned.

Yours sincerely,

            Dr. A. Marcell Szasz
            Semmelweis University
            Department of Internal Medicine and Oncology